# Expression of Potential Antibody–Drug Conjugate Targets in Cervical Cancer

**DOI:** 10.3390/cancers16091787

**Published:** 2024-05-06

**Authors:** Michael R. Mallmann, Sina Tamir, Katharina Alfter, Dominik Ratiu, Alexander Quaas, Christian M. Domroese

**Affiliations:** 1Faculty of Medicine, University of Cologne, 50931 Cologne, Germany; dominik.ratiu@uk-koeln.de (D.R.); alexander.quaas@uk-koeln.de (A.Q.); christian.domroese@uk-koeln.de (C.M.D.); 2Department of Obstetrics and Gynecology, University of Cologne, 50931 Cologne, Germany; 3Center for Integrated Oncology Aachen Bonn Cologne Düsseldorf, Germany; 4Department of Obstetrics and Gynecology, Hospital of the City of Cologne, 51069 Cologne, Germany; tamirs@kliniken-koeln.de (S.T.); alfterk@kliniken-koeln.de (K.A.); 5Department of Pathology, University of Cologne, 50931 Cologne, Germany

**Keywords:** antibody–drug conjugates, TROP2, CEACAM5, CD138, cervical cancer

## Abstract

**Simple Summary:**

With the introduction of antibody–drug conjugates (ADCs), novel treatment options for advanced and recurrent cervical cancer might be found. The expression of their target proteins in cervical cancer is largely unknown. This study addresses this issue by analyzing gene and protein expression of potential novel ADC targets in cervical cancer.

**Abstract:**

(1) Background: There is a huge unmet clinical need for novel treatment strategies in advanced and recurrent cervical cancer. Several cell membrane-bound molecules are up-regulated in cancer cells as compared to normal tissue and have revived interest with the introduction of antibody–drug conjugates (ADCs). (2) Methods: In this study, we characterize the expression of 10 potential ADC targets, TROP2, mesotheline, CEACAM5, DLL3, folate receptor alpha, guanylatcyclase, glycoprotein NMB, CD56, CD70 and CD138, on the gene expression level. Of these, the three ADC targets TROP2, CEACAM5 and CD138 were further analyzed on the protein level. (3) Results: TROP2 shows expression in 98.5% (66/67) of cervical cancer samples. CEACAM5 shows a stable gene expression profile and overall, 68.7% (46/67) of cervical cancer samples are CEACAM-positive with 34.3% (23/67) of cervical cancer samples showing at least moderate or high expression. Overall, 73.1% (49/67) of cervical cancer samples are CD138-positive with 38.8% (26/67) of cervical cancer samples showing at least moderate or high expression. (4) Conclusions: TROP2, CEACAM5 or CD138 do seem suitable for further clinical research and the data presented here might be used to guide further clinical trials with ADCs in advanced and recurrent cervical cancer patients.

## 1. Introduction

With an incidence of 600,000 new cases per year, cervical cancer represents the 4th most prevalent cancer type in women worldwide. Despite the introduction of vaccination and screening programs throughout the world, 340,000 deaths from cervical cancer occur worldwide per year [1]. The backbone of therapy in early cervical cancer remains the surgical approach, whereas in advanced cervical cancer, surgery has been replaced by chemoradiation. In case of recurrence or metastasis, chemotherapy combined with immunotherapy results in median progression-free survival rates of only a few months [2,3]. Consequently, additional therapeutic options after chemotherapy and immunotherapy are urgently needed.

The altered regulation of a variety of cellular components is a hallmark of cancer [4]. Among these cellular components are cell membrane-bound molecules that are up-regulated in cancer cells as compared to normal tissue. Many up-regulated membrane-bound proteins are known for many years yet interest has been revived in light of the introduction of antibody–drug conjugates (ADCs) [5]. ADCs represent new therapeutics in the landscape of oncological therapeutics. In ADCs, a highly potent chemotherapeutic agent is coupled to a monoclonal antibody. By combining a chemotherapeutic agent and an antibody together individually, a variety of target cell surface proteins on cancer cells can be targeted [5].

Currently, several ADC targets exist (TROP2, mesotheline, CEACAM5, DLL3, folate receptor alpha, guanylatcyclase, glycoprotein NMB, CD56, CD70 and CD138) with ADC compounds being tested in other cancer entities whose expression level in cervical cancer is of clinical interest.

With this study, we characterize the expression of potential ADC targets in both gene and protein expression in a well-defined cohort of cervical cancer patients.

## 2. Materials and Methods

Sixty-seven patients with the diagnosis of cervical cancer between 2010 and 2021 were included in this study as CCCC (Cologne Cervical Cancer Cohort) within the translational study GOProTAS (Gynecologic Oncology Protein Target Analysis Study; Ethics committee of the University of Cologne vote No. 22-1448). Surgical and medical treatment was performed according to the German clinical consensus guidelines. 

For gene expression analysis, publicly available transcriptomic data from the TCGA project were used. Data from 305 cervical squamous cell carcinoma samples were assessed using the UALCAN (The University of ALabama at Birmingham CANcer data analysis Portal) portal [6]. Briefly, the respective gene expression based on individual cancer stages was analyzed using the TCGA “cervical squamous cell carcinoma” dataset on the UALCAN portal. 

Immunohistochemistry (IHC) staining of tissue microarray was performed in 67 patients. Tissue microarrays (TMAs) of the 67 cervical cancer specimens were constructed as previously described [7]. Two intratumoral regions per case were identified on hematoxylin- and eosin-stained slides. Each region was sampled by four TMA cores with a diameter of 1.2 mm and an area of 1.13 mm^2^ as described before [8]. In brief, tissue cylinders with a diameter of 1.2 mm each were punched from selected FFPE tumor tissue blocks using a self-constructed semi-automated precision instrument and embedded in empty recipient paraffin blocks. Areas with alterations known to interfere with IHC were excluded, i.e., necrotic areas, fibrinous exudate, detritus and areas with artificial fragmentation. Protein expression of TROP2, CEACAM5 and CD138 was classified as IHC 0: no TROP2/ CEACAM5/ CD138 protein expression, IHC 1+: TROP2/CEACAM5/CD138 protein expression in <20% of tumor cells, IHC 2+: moderate TROP2/CEACAM5/CD138 protein expression in >20%, but <70% of tumor cells or IHC 3+: strong TROP2/CEACAM5/CD138 protein expression in >30% or moderate CEACAM5 expression in >70% of tumor cells. The following antibodies and staining protocols were used: TROP2 (abcam, clone ERP20043, rabbit monoclonal, 1:1000 EDTA buffer; CD138 (cellmarque, clone B-A38, mouse monoclonal, 1:100 EDTA buffer); CEACAM5 (dako, clone Il-7, mouse monoclonal, 1:200 citrate buffer). All slides were stained on Leica Bond Stainer, Wetzlar, Germany. All stains are routine immunohistochemical tests that are used in daily routine diagnostics and have therefore been repeated several times and are provided with suitable positive and negative on-slide controls.

Statistical analysis was performed using IBM SPPS Version 29 (IBM, New York, NY, USA), and figures were created using CorelDraw Graphics suite 2021 (Corel Corporation, Ottawa, ON, Canada). Chi Square tests were used where appropriate. All statistical analysis was two-sided and *p* values < 0.05 were considered to be significant.

## 3. Results

### 3.1. Clinicopathologic Parameters

For this study, 67 patients with the diagnosis of cervical cancer were included (Table 1). Mean age was 46.8 years (range 18–84 years). One half of patients were staged as stage I disease, 17.9% as stage II, 11.9% as stage III and 19.4% as stage IV. The majority of cervical cancer samples (67.2%) were of the squamous histologic subtype, 23.8% of the adenoid histologic subtype and 9% of the adenosquamous histologic subtype. Most cervical cancer samples were poorly (46.3%) or intermediately (50.7%) differentiated, and only 3% of the cervical cancer samples were highly differentiated. Furthermore, 49.3% of the cervical cancer samples showed lymphovascular space invasion; the majority did not express vascular space invasion (80.6%).

### 3.2. Overall Gene Expression of Potential ADC Targets in Cervical Cancer

To elucidate whether genes of potential ADC targets are expressed in cervical cancer tissue, we performed a gene expression analysis using the RNA-seq data of the TCGA data base on cervical cancer. 

We analyzed the overall gene expression of the 10 potential ADC targets TROP2, Mesotheline, CEACAM5, DLL3, folate receptor alpha, guanylatcyclase, glycoprotein NMB, CD56, CD70 and CD138. As can be seen in Figure 1, out of these 10 potential ADC candidates, only the three ADC targets TROP2, CEACAM5 and CD138 are expressed on the gene expression level in a reasonable amount to guide further protein analysis.

### 3.3. Gene and Protein Expression of TROP2 According to Tumor Stage in Cervical Cancer

To analyze gene expression of TROP2 over all tumor stages, gene expression of TROP2 in the TCGA dataset was analyzed according to FIGO stages I, II, III and IV (Figure 2A). TROP2 shows a reasonable strong gene expression in all tumor stages with no significant differences between the tumor stages. To elucidate whether this strong and continuous gene expression holds true on the protein level, we performed immunohistochemistry staining of TROP2 in the CCCC (Figure 2B; Table 2). Similar to the gene expression, TROP2 protein is strongly expressed over all tumor stages in cervical cancer. Overall, 98.5% (66/67) of cervical cancer samples are TROP2-positive with 94% (63/67) of cervical cancer samples showing at least moderate or high expression. TROP2 protein expression is statistically independent from the FIGO stage, histological subtype, grading, lymph vascular space invasion and vascular space invasion.

### 3.4. Gene and Protein Expression of CEACAM5 According to Tumor Stage in Cervical Cancer

We next checked for differences in CEACAM5 gene expression between different tumor stages (Figure 3A). 

CEACAM5 is far less expressed with respect to the transcript level in total, yet with similar gene expression levels over all tumor stages. In contrast to the stable gene expression profile, the proportion of patients with high or moderate protein expression of CEACAM5 declines from stage I over stage II to stage III and IV, yet these differences are not statistically significant. Overall, 68.7% (46/67) of cervical cancer samples are CEACAM-positive with 34.3% (23/67) of cervical cancer samples showing at least moderate or high expression (Figure 3B; Table 2). CEACAM5 protein expression is statistically independent from the FIGO stage, histological subtype, grading, lymph vascular space invasion and vascular space invasion.

### 3.5. Gene and Protein Expression of CD138 According to Tumor Stage in Cervical Cancer

We next analyzed gene and protein expression of the third potential candidate for ADC treatment in cervical cancer, CD138. 

Similar to the other two potential ADC candidates, CD138 shows a stable gene expression over all stages of cervical cancer with an intermediate strength of gene expression as compared to TROP2 and CEACAM5 (Figure 4A). Gene expression is relatively high in stage IV, an important aspect with respect to its use in the advanced and metastatic setting. This is comparable in protein analysis, where CD138 is rather low-expressed in tumor stages I and II, yet shows a pronounced expression especially in stage IV. Overall, 73.1% (49/67) of cervical cancer samples are CD138-positive with 38.8% (26/67) of cervical cancer samples showing at least moderate or high expression (Figure 4B; Table 2). CD138 protein expression is statistically independent from the FIGO stage, histological subtype, grading and lymph vascular space invasion, yet positively associated with vascular space invasion (*p* < 0.05) (Figure 5).

## 4. Discussion

Considering the low rates of progression-free survival and overall survival, there is a huge unmet clinical need for novel treatment strategies in advanced and recurrent cervical cancer that are both effective and well tolerable. A treatment that specifically targets a tumor cell by recognizing its individual tumor cell-specific surface molecules offers a more precise treatment option than the currently available single chemotherapy regimen with a potential better efficacy–toxicity profile. Nevertheless, efficacy of the same chemotherapeutic agent differs tremendously between tumor types and the clinical benefit of ADCs in other cancer entities does not prove its efficacy in cervical cancer. Consequently, experimental research is urgently warranted to delineate whether novel ADCs represent an effective treatment strategy for recurrent, progressive or metastasized cervical cancer in general, identifying which patients profit most from its use and which side effects must be expected. ADCs treatment has been shown to be effective if their targets are expressed in cervical cancer. One of these ADC targets in cervical cancer is the tissue factor. The tissue factor is highly expressed in cervical carcinoma and can be targeted by the ADC tisotumab vedotin. Treatment with tisotumab vedotin resulted in clinically meaningful antitumor activity with a manageable safety profile encouraging its use in this tumor entity and strengthening the potential of ADCs treatment in cervical cancer [9,10]. In line with these results, our study adds to the knowledge of potential ADCs targets in cervical cancer. 

Mesotheline is a glycosylphosphatidylinositol (GPI)-anchored cell surface glycoprotein with limited expression in normal tissue yet overexpression in many cancer tissues [11]. Gene expression of mesotheline has been described in a variety of adenocarcinomas such as ovarian cancer, pancreatic adenocarcinoma, endometrial carcinoma, and adenocarcinoma of the lung and mesotheliomas, yet expression in squamous cancers such as squamous cancers of the cervix, the anal canal, the skin, the esophagus or the vulva is limited, an observation that is in line with our observation of missing expression in squamous cervical cancer [12,13]. 

DLL3 represents one of the ligands of the notch signaling pathway and is expressed in several tumor entities including small lung cancer and neuroendocrine tumors [14]. The ADC rovalpituzumab tesirine targets the DLL3 protein [15,16,17,18]. There are currently no convincing analyses of DLL3 expression in cervical cancer.

Folate receptor alpha is a glycosylphosphatidylinositol-anchored membrane protein responsible for folate uptake of cells and is highly expressed in several epithelial solid tumors, such as endometrial cancer, ovarian cancer, colorectal cancer, triple-negative breast cancer and lung cancer [19,20]. Clinical trials with the ADC mirvetuximab soravtansine have shown promising clinical efficacy in ovarian cancer [21]. Folate receptor alpha expression has been associated with cervical carcinogenesis and has been shown to be present in a subset of cervical cancer patients [22,23].

Guanylatcyclase is a member of the guanylyl cyclase (GC) family of proteins that are involved in regulation of intracellular cGMP concentrations [24]. Guanylatcyclase expression has so far been mainly attributed to gastrointestinal malignancies such as gastric and gastroesophageal junction cancer, pancreatic cancer and colon cancer, but clinical trials with the ADC TAK-264 have shown only limited clinical efficacy [24,25,26].

Glycoprotein NMB (GPNMB) expression has been described in melanoma, glioma and breast cancer [27,28,29,30]. The ADC glembatumumab vedotin (CDX-011) has been investigated in clinical trials for breast cancer with limited clinical benefit compared to current standard therapy [31].

CD56 or NCAM (neural cell adhesion molecule) is widely expressed in a variety of mainly pediatric tumors such as neuroblastoma, Wilms tumor and acute myeloid leukemia. Only sparse data exist on the expression of CD56 in cervical cancer; overall, CD56 seems to be progressively down-regulated throughout the carcinogenesis of cervical cancer [32]. CD56 can be targeted by the ADC lorvotuzumab mertansine that has shown only limited efficacy in a phase 2 study in children with relapsed or refractory Wilms tumor, rhabdomyosarcoma, neuroblastoma, pleuropulmonary blastoma, malignant peripheral nerve sheath tumor or synovial sarcoma [33,34]. 

CD70 is primarily expressed on immune cells and is involved in the phosphatidylinositol-3 kinase (PI3K) and MAP kinase signaling pathways [35]. Overexpression of CD70 has been associated with a variety of hematologic malignancies and solid tumors [36,37,38].

Vorsetuzumab mafodotin is the monoclonal antibody vorsetuzumab conjugated with monomethyl auristatin FADC (MMAF) that started clinical trials for renal cell carcinoma yet has not shown convincing clinical activity [39].

CD138 is a molecule whose expression has so far been attributed mainly to multiple myeloma. Indatuximab ravtansine (BT062) is an antibody–drug conjugate that binds to CD138 and has shown preliminary antitumor activity in refractory multiple myeloma [40].

These aforementioned potential ADCs’ candidates mesotheline, DLL3, folate receptor alpha, guanylatcyclase, glycoprotein NMB, CD56 and CD70 whose targets are expressed in other cancer entities could not be considered for further protein expression analysis due to missing gene expression. Consequently, these potential ADC target genes do seem suitable for further clinical research in cervical cancer. 

Among the three potential candidates TROP2, CEACAM5 and CD138 that were analyzed further, TROP2 shows the most promising protein expression as it shows a high protein expression over all tumor stages and might even be an interesting therapeutic candidate for earlier stages. The trophoblast cell surface antigen 2 (TROP2) is a transmembrane calcium signal transducer that is involved in multiple intracellular signaling pathways such as the MAPK and the PI3K/AKT pathway [41]. With these pathways involved in proliferation, migration and invasion of cancer cells, TROP2 expression is up-regulated in the majority of tumor types such as breast, colon, lung, ovary, rectum and endometrium [41,42]. TROP2-directed ADCs have been among the first to be introduced in clinical practice. Sacituzumab Govitecan (TRODELVY^®^) is a TROP2-directed ADC with the TROP2-directed monoclonal antibody sacituzumab coupled to the topoisomerase I inhibitor SN-38 (an active metabolite of Irinotecan) [43]. Preclinical data from cervical cancer cell lines suggest a high efficacy of sacituzumab govitecan in the treatment of cervical cancer [44]. This is in line with our results that show a consistently high expression of TROP2 in cervical cancer. The high expression of the TROP2 membrane protein in stage IV cervical cancer is deemed suitable for TROP2-directed treatment approaches without prior protein staining of tumor samples. Sacituzumab govitecan, a TROP2-directed ADC, has shown promising clinical efficacy in patients with diverse types of cancers and both the U.S. Food and Drug Administration (FDA) and the European Medicines Agency (EMA) granted sacituzumab govitecan accelerated approval for the treatment of metastatic triple-negative breast cancer [45,46]. The phase I/II IMMU-132-01 basket trial includes a variety of epithelial cancer types. Among those, there is only one patient with cervical cancer included; consequently, no clinical data exist on the efficacy of sacituzumab govitecan in cervical cancer [47]. Due to the high expression of TROP2 in our cervical cancer cohort, sacituzumab govitecan might be an interesting compound for a TROP2-directed phase III trial in advanced and recurrent cervical cancer. 

The CEACAM5 gene, a member of the CEACAM gene family, encodes CEA (carcinoembryonic antigen) and represents a widely used tumor marker in various cancer types such as colorectal, gastric, pancreatic, lung, ovarian and endometrial cancer [48]. As expression of CEACAM5 in normal tissue is limited, it represents a promising target for CEACAM5-directed therapeutic approaches. The ADC target protein CEACAM5 showed tumor gene and protein expression mainly confined to the early stages I and II. Nevertheless, a minority of patients show high CEACAM5 gene expression also in the higher stages, although not represented in our protein staining assay, which might confine anti-CEACAM5 treatment exclusively to patients with prior CEACAM5 staining of their tumor tissue. Tusamitamab ravtansine (SAR408701) is an anti-CEACAM5-DM4 antibody–drug conjugate currently investigated in clinical trials for its safety and antitumor activity in patients with advanced solid tumors [49]. Only recently, the pharma company Sanofi has announced to discontinue the global clinical development program of tusamitamab ravtansine based an interim analysis of the Phase 3 CARMEN-LC03 trial evaluating tusamitamab ravtansine as monotherapy compared to docetaxel in previously treated patients with metastatic non-squamous non-small cell lung cancer (NSCLC) whose tumors express high levels of CEACAM5. Tusamitamab ravtansine as a monotherapy did not meet its primary endpoint of progression-free survival (PFS) compared to docetaxel. Labetuzumab govitecan represents an anti-CEACAM5 antibody conjugated to 7-ethyl-10-hydroxycamptothecin (SN-38) tested for its safety and antitumor activity in patients with metastatic colorectal cancer [50].

CD138 is a molecule whose expression has so far been attributed mainly to multiple myeloma. CD138 represents a promising candidate for a potential ADC treatment in cervical cancer patients, as a substantial part of samples show a high protein content. Despite its gene expression in both local and advanced stages, CD138 was deemed inferior to TROP2 due to the higher gene and protein expression of the latter in both local and advanced stages of cervical cancer. To our knowledge, the ADC indatuximab ravtansine (BT062) binds to CD138 and has so far only been investigated in refractory multiple myeloma. The expression of CD138 in cervical cancer might be an interesting target for clinical trials investigating the role of indatuximab ravtansine in the treatment of advanced or recurrent cervical cancer patients.

Overall, at least three potential ADC targets exist in cervical cancer that warrant further clinical investigation. However, the high rate of studies that lead to a discontinuation of the further clinical development of ADCs might limit the initial euphoria about the general effectiveness of ADCs in solid tumors. In consequence, the exact role of ADCs in cancer treatment remains to be established for each cancer entity. As most ADCs have so far been investigated as single agents only, a combination of several ADCs or a combination of ADCs with immunotherapy or other chemotherapeutic compounds might increase clinical efficacy in solid tumors. Yet as long as the exact mechanisms that cause resistance to these highly specific cancer drugs are unknown, more preclinical research on the resistance mechanisms against ADCs should also be performed.

## 5. Conclusions

In conclusion, we present a comprehensive analysis of potential ADC targets in cervical cancer. The data presented here might be used to guide further clinical trials with ADCs in advanced and recurrent cervical cancer patients.

## Figures and Tables

**Figure 1 cancers-16-01787-f001:**
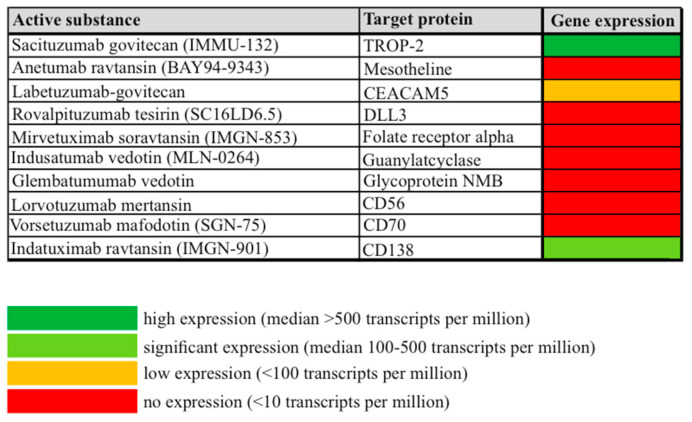
Overall gene expression of 10 potential ADC target proteins in the TCGA dataset of cervical cancer. Gene expression was divided into high expression (>500 specific gene transcripts per million transcripts), significant expression (100–500 specific gene transcripts per million transcripts), low expression (<100 specific gene transcripts per million transcripts) and no expression (<10 specific gene transcripts per million transcripts).

**Figure 2 cancers-16-01787-f002:**
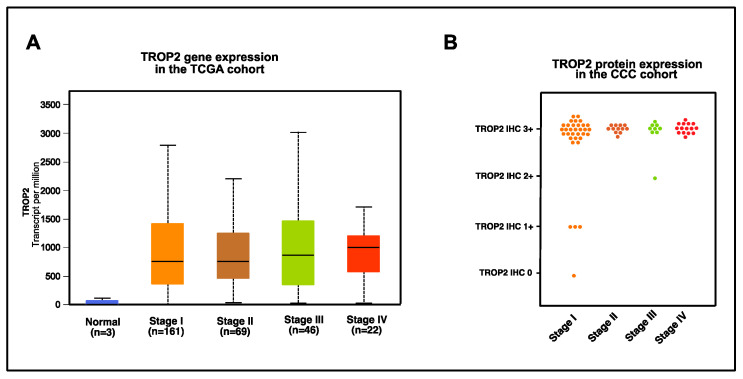
(**A**) TROP2 gene expression in the TCGA cohort. Gene expression of TROP2 in the TCGA dataset in cervical cancer according to age, visualized as transcripts of CEACAM5 per million transcripts overall with median, interquartile range according to tumor stage and normal tissue and (**B**) TROP2 protein expression in the CCC cohort, visualized as IHC 0 to 3+ according to cervical cancer tumor stage.

**Figure 3 cancers-16-01787-f003:**
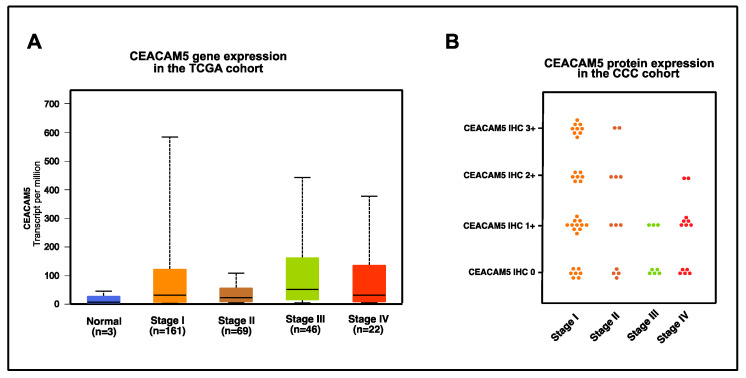
(**A**) CEACAM5 gene expression in the TCGA cohort. Gene expression of CEACAM5 in the TCGA dataset in cervical cancer according to age, visualized as transcripts of CEACAM5 per million transcripts overall with median, interquartile range according to tumor stage and normal tissue and (**B**) CEACAM5 protein expression in the CCC cohort, visualized as IHC 0 to 3+ according to cervical cancer tumor stage.

**Figure 4 cancers-16-01787-f004:**
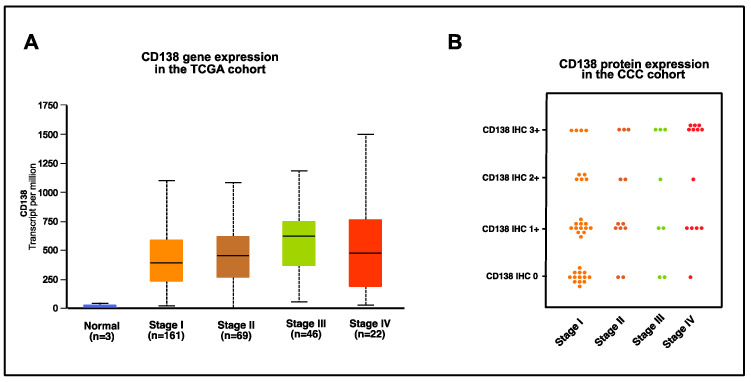
(**A**) CD138 gene expression in the TCGA cohort. Gene expression of CD138 in the TCGA dataset in cervical cancer according to age, visualized as transcripts of CD138 per million transcripts overall with median, interquartile range according to tumor stage and normal tissue and (**B**) CD138 protein expression in the CCC cohort, visualized as IHC 0 to 3+ according to cervical cancer tumor stage.

**Figure 5 cancers-16-01787-f005:**
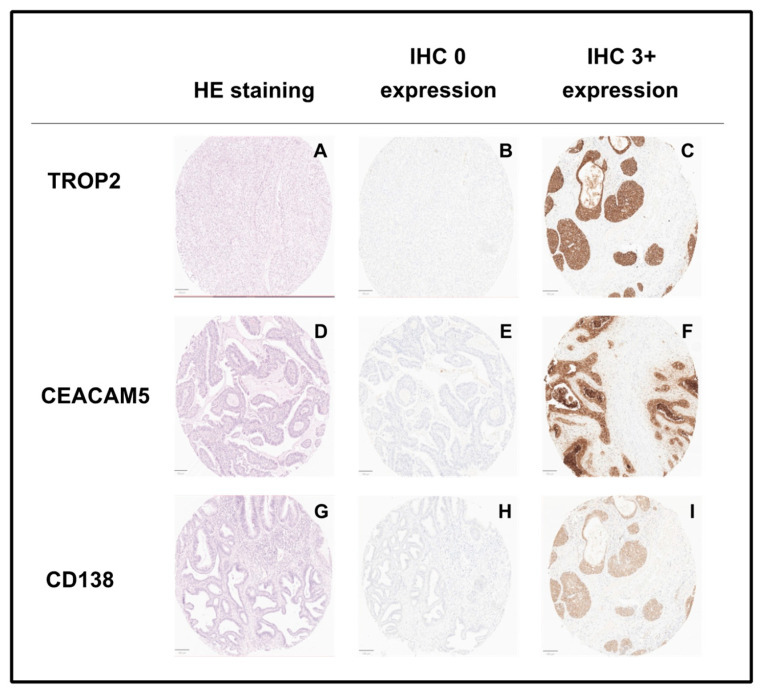
Representative images of tissue microarray TMA staining revealing hematoxylin eosin (HE) staining (**A**,**D**,**G**), missing expression of TROP2 (**B**), CEACAM5 (**E**) and CD138 (**H**) and high expression of TROP2 (**C**), CEACAM5 (**F**) and CD138 (**I**); objective amplification ×100.

**Table 1 cancers-16-01787-t001:** Clinicopathological characteristics of 67 patients with cervical cancer.

Clinicopathologic Characteristics		Title 3
Mean age at diagnosis (years)		47.5 (range, 18–84)
Tumor stage (FIGO classification)		
	Stage I	34/67 (50.7%)
	Stage II	12/67 (17.9%)
	Stage III	8/67 (11.9%)
	Stage IV	13/67 (19.4%)
Histologic subtype		
	squamous	45/67 (67.2%)
	adenocarcinoma	16/67 (23.8%)
	adenosquamous	6/67 (9%)
Grading		
	G1	2/67 (3%)
	G2	34/67 (50.7%)
	G3	31/67 (46.3%)
Lymphovascular space invasion		
	L0	34/67 (50.7%)
	L1	33/67 (49.3%)
Vascular space invasion		
	V0	54/67 (80.6%)
	V1	13/67 (19.4%)

**Table 2 cancers-16-01787-t002:** ADC target protein expression in 67 patients with cervical cancer.

Protein Expression		
TROP2		
	TROP2 IHC 0	1.5% (1/67)
	TROP2 IHC 1+	4.5% (3/67)
	TROP2 IHC 2+	1.5% (1/67)
	TROP2 IHC 3+	92.5% (62/67)
CEACAM5		
	CEACAM5 IHC 0	31.3% (21/67)
	CEACAM5 IHC 1+	34.3% (23/67)
	CEACAM5 IHC 2+	17.9% (12/67)
	CEACAM5 IHC 3+	16.4% (11/67)
CD138		
	CD138 IHC 0	26.9% (18/67)
	CD138 IHC 1+	34.3% (23/67)
	CD138 IHC 2+	13.4% (9/67)
	CD138 IHC 3+	25.4% (17/67)

## Data Availability

The raw data supporting the conclusions of this article will be made available by the authors on request.

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
