# Peer review of "Expression of Potential Antibody–Drug Conjugate Targets in Cervical Cancer"

_cancers, 2024, doi:10.3390/cancers16091787_

Round 1
Reviewer 1 Report
Comments and Suggestions for Authors
The authors present an overview and gene expression analysis for potential ADC targets in cervical cancer. The paper is well written and identifies potential targets for further investigation.
I have the following comments/questions.
1. The introduction is very lengthy; consider shortening and moving some information to the discussion.
2. The methods should be a little more descriptive. Was a control used? Did the authors perform repeat testing to confirm accuracy?
3. In table 1, 'stadium' is used which is not typical ('stage' recommended).
4. The discussion does not mention an ADC that has phase III data, tisotumab vedotin. I think this must be included for completeness.
Reviewer 2 Report
Comments and Suggestions for Authors
This paper presents an interesting study on the expression of potential antibody-drug conjugate (ADC) targets in cervical cancer. While the findings contribute valuable insights into the field, I suggest the authors consider investigating and discussing the antigen expression levels using IHC (Immunohistochemistry) scoring, such as the IHC score or H score. This approach could provide a more quantitative and standardized evaluation of antigen expression, thereby enriching the discussion and implications of the study findings.
Round 2
Reviewer 2 Report
Comments and Suggestions for Authors
No further revisions are needed.
